# Urban water systems: Development of micro-level indicators to support integrated policy

**Olivia Jensen**[1,2]*, **Adilah Khalis**[1]

**1** LRF Institute for the Public Understanding of Risk, National University of Singapore, Singapore, Singapore, **2** Institute of Water Policy, Lee Kuan Yew School of Public Policy, National University of Singapore, Singapore, Singapore

* olivia.jensen@nus.edu.sg

**Data Availability Statement:** The data have been deposited in the repository Open ICPSR https://www.openicpsr.org/openicpsr/project/117501/version/V1/view/

## Abstract

Urban water systems involve complex interactions between ecological, social and economic factors. Integrated management approaches are needed to achieve multiple policy objectives in the sector and can be pursued at a range of spatial scales. Small-scale integrated water projects are both feasible and valuable in dynamic urban environments in developing countries. This paper develops a method for the prioritization of localities for integrated projects and applies this to the city of Jakarta. A set of indicators is defined following a systems approach, populated, displayed through a dashboard and mapped, and the relationships between indicators are analysed. Indicator-based prioritization allows policy-makers to guide resources to integrated projects to contribute effectively to the achievement of policy goals.

## 1 Introduction

Urban water systems encompass ecological, social and economic factors. Within these systems, natural water resources and ecosystems interlink with infrastructure for water supply, collection and treatment of wastewater and flood protection. These interact with the behavior of people, firms and governments in their use of water for health, recreation, livelihoods and economic activities.

The multi-faceted nature of the urban water system is reflected in the wide-ranging set of policy goals relevant to the sector. This is illustrated by the Sustainable Development Goals (SDGs) for water, which cover access to water supply and sanitation, water pollution, resource conservation, ecosystem restoration and integrated management. Many governments have additional policy objectives relating to flood risk management, energy use, service quality and public participation in decision-making. The interconnections between aspects of the sector imply that interventions designed to meet one policy objective may have unintended positive or negative consequences for the achievement of other objectives.

Urban water systems are subject to increasing uncertainty as a result of rapid urbanization and densification of built-up areas, economic development, changes in climate and interconnections with energy and food systems. Policies and management strategies which were effective in meeting policy goals in the past, like centrally operated distribution and treatment

**Funding:** The authors gratefully acknowledge funding support from the Institute for the Public Understanding of Risk, National University of Singapore under grant R-727-003-003-133 and the Institute of Data Science, National University of Singapore (http://ids.nus.edu.sg) under the WATCHA: WATer CHallenges Analytics grant R-252-000-650-646. The funders played no role in the study design, data collection and analysis, decision to publish or preparation of the manuscript.

**Competing interests:** The authors have declared that no competing interests exist.

systems, may no longer be able to cope with the scale and dynamic nature of contemporary challenges. These pressures are likely to be even greater in high-growth cities in developing countries where existing infrastructure does not provide universal access to safe water and sanitation.

The design of appropriate interventions to achieve water policy objectives within this complex system requires a system-level approach like that of integrated water resources management (IWRM). IWRM is a well-established framework in the water sector which takes into account both human and ecological needs, and can be defined as "a process which promotes the coordinated development and management of water, land and related resources, in order to maximise the resultant economic and social welfare in an equitable manner without compromising the sustainability of vital eco-systems." [1]. IWRM is endorsed by many national governments across regions and levels of economic development and at the global level through its inclusion in the SDGs.

At the city level, a range of concepts have been proposed for the application of IWRM principles, including Integrated or Sustainable Urban Water Management (IUWM) [2–6], Total Water Cycle Management [7], Water Sensitive Urban Design or Cities [8,9]. While these concepts have different emphases, they all reflect a shift from traditional, centralised engineering-focused management towards approaches which take into account system-level interlinkages and user preferences [10].

In the context of IUWM, increasing attention is being given to the potential for small-scale distributed systems to complement the centralized network. These can combine water, wastewater and solid waste treatment, reducing network costs and providing economies of scope, such as co-treatment of organic waste and wastewater and the direct use of biogas. Distributed infrastructure may provide greater flexibility to respond to changing conditions, reduce risk and contain the impact of failures, reduce costs associated with transmission and distribution, strengthen local communities and economies and allow for more sensitivity to local conditions and impacts [11–13].

The benefits of distributed systems may be particularly high in cities in developing country cities where urban water systems are highly fragmented in terms of sources, technologies and actors, leading to poor and unequal outcomes [14,15]. Households in these cities are obliged to patch together water supply for different uses from a range of sources, sometimes leading to the unsustainable use of local water resources [10]. However, the failure of existing models in these challenging urban contexts may provide the opportunity and incentives for transitions to integrated water governance and management [16]. Studies in Vietnam and China [15,17] point to significant potential benefits from integrated projects in expanding Asian cities. If designed appropriately, IUWM projects can contribute to multiple policy goals while avoiding unintended effects of policies designed to tackle a single policy objective [18].

Despite the potential benefits, mainstreaming IUWM has often proved challenging [4]. Governance structures and embedded interests can restrict incentives to innovate and the costs of retro-fitting existing systems may be prohibitive, limiting IUWM interventions to distributed systems in new build areas [19]. Further challenges include interactions between decentralised projects and existing centralised infrastructure and whether the projects can be economic and ecologically sustainable in the long-term [18] as well as the limited implementation capacities of the sector [20].

Currently, the selection of sites for IUWM projects is often ad hoc and opportunistic. While the ad hoc approach may sometimes offer advantages, as it is able to capitalize on leadership and community motivation at the micro-level, it is unlikely to be optimal in terms of efficiency or equity when considered from the perspective of the achievement of policy goals. Local interventions need to be aligned and coordinated by a strong strategy at the city level

and implemented using consistent methods in order to maximize the contribution of IUWM to meeting policy objectives [21]. An evidence base is needed to inform such a strategy.

The objective of this paper is to develop an evidence base for IUWM strategy for the city of Jakarta, Indonesia, to assist government agencies, utilities and financial institutions in prioritizing projects for funding and monitoring implementation in the context of limited resources. The paper aims to contribute to the growing literature on IUWM and evidence-based approaches to project selection and evaluation through the development of micro-level indicators for the water sector. In policy terms, the paper seeks to support the take-up of IUWM approaches in Indonesia, using Jakarta as a demonstration case, with potential application to other countries.

Jakarta provides an interesting setting within which to study IUWM adoption as a confluence of factors opened a window for a transition to IUWM in Jakarta in the late 2010s. The central government adopted an ambitious target to expand access to safe water supply to 100% by 2024 and the local government set an additional target to expand piped supply to 100% by 2030. However, the local government faces budget constraints and restrictions on raw water availability. Surface waters in Jakarta are highly polluted and efforts to secure additional raw water supplies from outside the city have been unsuccessful; groundwater has been over-exploited, contributing to land subsidence and saltwater intrusion. Local government agencies and the city's private concessionaires are therefore experimenting with alternative ways to expand supply through IUWM, with the support of the World Bank, the Association of Indonesian Municipal Governments (APEKSI) and the central government.

Within this policy context, this paper develops and analyses a set of micro-level indicators to measure the performance of the urban water system across Jakarta using the frame of water security. The approach and method for the selection and population of the indicators are set out in Section 2. Section 3 introduces the study area in more detail. Section 4 highlights findings on the relationships between indicators which are discussed in Section 5. Section 6 concludes and proposes steps for further research.

## 2 Approach & method

Our objective is to develop a systematic basis for prioritization of localities for IUWM interventions in Jakarta within the context of highly differentiated performance and dynamic change. Our unit of analysis is the smallest urban administrative jurisdiction in Indonesia, known as "kelurahan" or village, which corresponds to the appropriate scale for local IUWM projects indicated in the literature reviewed.

Our starting point is to develop a set of indicators to measure the current attributes of the water system. We frame performance in terms of water security, which we interpret broadly, following the UN definition of water security as, "The capacity of a population to safeguard sustainable access to adequate quantities of acceptable quality water for sustaining livelihoods, human well-being, and socio-economic development, for ensuring protection against water-borne pollution and water-related disasters, and for preserving ecosystems in a climate of peace and political stability." [22]. We understand water security as an over-arching policy objective which encompasses objectives articulated in the Sustainable Development Goals as well as the management of flooding and other water-related risks.

We adopt a systems approach to the development of indicators as urban water security is affected by many interrelated mechanisms resulting in a high degree of complexity and the systems approach can help to provide clarity on these interactions and the underlying causal relationships [23–25]. We focus on individual indicators and how they interact in the context of IUWM interventions, using a dashboard approach and calculating a simple unweighted index.

Further work could be done to refine this into a composite index by prioritizing indicators and establishing weights through multi-criteria decision analysis (MCDA) [26].

Following the work of van Ginkel et al [27], we employ the Pressure-State-Impact-Response (PSIR) framework to develop indicators. PSIR is a well-established approach for the development of indicators in dynamic environmental systems [28–31] and has been applied to water-related issues [32–35].

In this framework, pressures are factors which influence the state of the system. These are sub-divided into environmental and socioeconomic categories. Environmental pressures include characteristics of the climate and hydrology. Socioeconomic pressures relate to demographics, characteristics of spatial development and economic activity. State refers to the current properties of the system, either natural or built, including infrastructure for the collection and treatment of water and flood protection. We include variables on the extent and quality of network service provision. Impact refers to outcomes understood in terms of the functions of the system, from the point of view of the citizen, reflected in access to safe water and disease incidence, and of the environment, reflected in resource degradation. Responses refer to actions taken by policymakers, firms and households in relation to water services. These are captured through qualitative assessments of water policy and strategy.

Using this framework, we develop a set of indicators. These are summarized in Table 1.

The indicators were developed iteratively. First, we identified a preliminary set of 54 pressure, state, impact and response indicators based on the framework and literature. 25 of these are measurable only at the city level and not at higher spatial resolution. These city-level indicators are discussed in Section 3. For the purposes of the indicator set, we focus on the 25 indicators for which there is variation between the micro-level administrative units ("kelurahan") or villages.

The next step was to populate the indicators. The sources of data, units, scaling and additional remarks are provided as supplementary information for the paper. Where possible, we used publicly available data from official government sources. The main sources of data were census data from the national statistical agency (Badan Pusat Statistik, BPS) and data from the Jakarta statistical agency. Data on piped connections per micro area or village were provided by the water utility, Pam Jaya.

Where data were not available, we identified a suitable proxy. For example, micro-level estimates of economic activity are not available from the statistical agencies. We therefore construct a variable based on Night-Time Lights data. A full explanation of the construction of this variable is provided in the supplementary information. For flooding, the publicly available data do not distinguish between riverine, coastal and stormwater flood events so we combined these into a single flood incidence indicator. Where no data were available at the desired spatial scale, we were obliged to drop the indicator from the set. The database format allows for the indicator set to be updated if more information becomes available. Following this process, we were left with a set of 17 indicators.

A dashboard interface for interrogation and display of indicator data was developed in Microsoft Excel. The dashboard approach allows interested parties to compare administrative units on a single dimension or to view data for all indicators for a single administrative unit. The presentation of the data in this accessible format is intended to facilitate its use by decision-makers as well as other researchers. Access to the dashboard is available from the authors on request.

We then transform all the indicators into a 5-point scale and aggregate the total into an unweighted water security score for each administrative unit. This simple method of aggregation weights all constituent indicators equally. Policymakers and other interested parties may wish to apply different weights to the constituent elements, which is facilitated by the

**Table 1. Indicators selected for the urban water security index for Jakarta using the PSIR framework.**

| CODE* | INDICATOR | METRIC | RANGE/SCALE |
|---|---|---|---|
| 1000 | **PRESSURE INDEX** | | |
| 1100 | **Environmental pressures** | | |
| 1104 | Elevation | Elevation above sea-level | -5 to 44m |
| 1200 | **Socioeconomic pressures** | | |
| 1201 | Population growth | Annual population growth | % |
| 1202 | Slums | Slum density | % |
| 1203 | Economic activity | Night-time light radiance | 10 to 120 |
| 1204 | Non-domestic demand | Water usage of small-medium industries | 0–2 |
| 1205 | Industrial activity | Industrial zones | Binary (0/1) |
| 2000 | **STATE INDEX** | | |
| 2100 | **Water Service** | | |
| 2101 | Piped water access | Piped water network coverage | % |
| 2102 | Piped water pressure | Piped water pressure (percentage of months in a year with low water pressure) | 5 categories |
| 2200 | **Water Quality** | | |
| 2201 | Drinking water quality | City-level | - |
| 2202 | Groundwater quality | Groundwater conservation zone classification | 1 to 3 |
| 2300 | **Infrastructure** | | |
| 2301 | Wastewater disposal | Population with access to septic tank | % |
| 2400 | Flood protection infrastructure | City-level (qualitative) | - |
| 3000 | **IMPACT INDEX** | | |
| 3100 | **Water Supply** | | |
| 3101 | Access to safe water | Population using protected water sources | % |
| 3102 | Reliance on groundwater | Groundwater consumption (litres per capita per day) | 0 to 10 |
| 3200 | **Health** | | |
| 3201 | Sanitation access | Population with access to toilet | % |
| 3202 | Waterborne disease risk | Diarrhoea prevalence rate (no. of cases per 10,000 people) | 0 to 950 |
| 3203 | Water-related disease risk | Dengue prevalence rate (no. of cases per 10,000 people) | 0 to 17 |
| 3300 | **Environment** | | |
| 3301 | Groundwater over-exploitation | Change in Groundwater Conservation Zone (2013–2017) | -1 to 1 |
| 3302 | Flood incidence | Number of years flooded between 2013 to 2016 | 0 to 4 |
| 4000 | **RESPONSE INDEX** | City-level (qualitative) | |

*Codes are not consecutive as city-level indicators have been excluded from the table.

dashboard interface. A consistent set of weights for all stakeholders could be established through a MCDA approach.

In October 2019, focus group discussions were conducted in Jakarta, focusing on the identification of suitable IUWM models. Five discussions with 4–8 participants per group were held. Participants were purposively selected to represent central and local government departments responsible for water resource management and water service provision, sector associations and financiers. The focus group findings are reported briefly in Section 6.

## 3 Study area: Jakarta

This section presents the interlinkages between water resources, infrastructure, policy and governance of the water system in Jakarta. Overall, Jakarta faces a high level of water risk associated with the limited coverage and quality of piped water supply, poor sanitation, pollution of

surface water sources, over-abstraction of groundwater, land subsidence and high riverine, pluvial and coastal flood risks.

The special capital region of Jakarta (DKI Jakarta) has a population of 10.64 million. It forms the central part of a larger metro area of more than 30 million people known as Jabodetabek. Jakarta's population has grown by 27% since 1990 and continues to grow at a rate of around 1.1% a year.

Land elevation falls from south to north, with the most densely built areas of the city found in the downstream area. 13 rivers and 2 canals flow across Jakarta from south to north and have estuaries in the Java sea along a stretch of coastline of approximately 35 km. Jakarta's average annual rainfall is 1816mm with monthly variation of 43-300mm. There is evidence of increasing rainfall extremes which contributed to very severe floods in 2007 and 2013 [36].

Piped water supply coverage in Jakarta is far from universal. The public water service agency, Pam Jaya, estimated coverage to be 73% in 2017. Of those households which do have a piped connection, many receive intermittent supply: 45% of customers in the western area of the city and 62% in the eastern area have 24-hour service [37]. Water pressure is also highly variable across the city. Less than half the water supplied meets the service standard of 0.75atm [37].

Piped network supply in Jakarta is constrained in part due to the limited availability of raw water. Around 80% of the city's water supply is drawn from the Jatiluhur Dam in the neighbouring province of Bekasi. The allocation of water from the dam is set under the authority of a state-owned company, PJT2, and the allocation has not been increased since 1997. The allocation is equivalent to less than half the estimated water demand of the city [38]. The quality of water in the dam has declined in recent years and the number of competing users of dam water has risen.

Supplies from Jatiluhur are supplemented by bulk treated water purchased from the neighbouring province of Tangerang. In future, a new dam is planned at Karian in Banten province but the timeframe for this is not certain [39]. Low raw water availability is exacerbated by the poor condition of the water transfer and distribution infrastructure in Jakarta. Non-Revenue Water (NRW) was estimated at 44.16% in 2017 due to physical leaks and unauthorized connections.

Low-income households not connected to the network sometimes buy piped water from neighbours. The per-unit price of purchases from neighbours varies widely, averaging more than six times the cost of piped water through a formal connection [40].

Households at a range of income levels use groundwater for household purposes to supplement or replace piped water. High-rise apartment blocks catering to higher income groups generally draw groundwater from the confined deep aquifer while other households draw water through shallow wells from the unconfined shallow aquifer [37,40]. Domestic use of groundwater from the shallow layer is allowed without a permit (except for 'affluent households') and no abstraction fees are payable. Government institutions are also able to use groundwater without a permit or fee. Commercial and industrial users are required by regulations to register borewells and monitor abstractions, but many do not do so. Since 1998, a fee has been imposed for the abstraction of groundwater, but it has been patchily enforced. The northern part of the city area has been designated a zero-abstraction area in which no new deep wells are authorized, yet unauthorized abstraction continues.

Neither piped water nor groundwater is usually potable and households generally rely on bottled water for drinking purposes when they can afford to do so.

Over-withdrawals from the contained aquifer have led to salinization of the shallow subsurface layer [41,42]. Availability and quality of groundwater are expected to worsen further over time [43].

Intensive abstraction of groundwater has also contributed to land subsidence, along with natural consolidation of alluvial soil and settlement of high compressibility soil due to construction [41]. Between 1974 and 2010, land subsidence in Jakarta typically varied from 3–10 cm/year across the city, with cumulative subsidence of 4 metres in some areas over this period. The impact of subsidence is seen in damage to housing, buildings and infrastructure, changes in river canal and drain flow systems, increased inland sea water intrusion and perhaps most significantly, wider and more severe flooding.

Jakarta has a long history of seasonal flooding during monsoons but in recent years flooding appears to have become more frequent and affected larger areas of the city. The floods of 2007 and 2013 were the most destructive recorded. In 2007, 75% of the city was flooded and 430,000 people were displaced. Damage to infrastructure and assets was estimated at US$900 million [44]. In 2013, the breach of the western flood canal dike resulted in 10–20 days of severe flooding in the northern areas of the city, while floods in greater Jakarta in early 2020 led to more than 60 deaths.

Since the floods of 2007, considerable efforts have been made by the Jakarta government to improve flood protection for the city, with the support of the World Bank. The Eastern Banjir (Flood) Canal was constructed and existing canal system has been dredged, renewed and extended [45]. To address coastal flood risk, the national and city governments adopted a master plan in 2014. The first phase, extending and strengthening the current sea wall, has been completed. Subsequent phases which include the construction of an outer sea wall defence, are under evaluation. While these measures have effectively reduced flood risk in some locations in the city in the short-term, risks are expected to rise in the future as a result of sea-level rise, subsidence and ongoing development and land use changes in upstream areas.

Sanitation coverage is extremely limited in Jakarta. Jakarta has only one functional wastewater treatment plant which has a capacity of 22 million litres per day (MLD), capable of treating less than 5% of the wastewater produced by the city [38]. The majority of households have septic tanks for the disposal of wastewater.

The very low level of wastewater collection and treatment has contributed to high levels of contamination in environmental waters and potentially irreversible pollution of surface waters and shallow aquifers [46]. Dsikowitzky [47] estimates that 5–17 tons of pollutants from municipal sources are carried by just one urban river, the Ciliwung, into Jakarta Bay each year. The wide distribution of fecal contamination in Jakarta Bay is also a concern for food safety in aquaculture and local fisheries.

Turning to the policy context, Indonesia is committed to the SDGs. In addition, the national government has set a target of universal coverage to safe water supply by 2024. With development partners, strategies have been developed to increase raw water supply [39] and sanitation [48] but the government has not committed to timelines for implementation.

Currently, governance of the water sector is Jakarta is complex and highly fragmented. Water supply, water resource management, groundwater, wastewater and flood management are all under the responsibility of different government departments. In addition, Indonesia has a decentralised mode of government under which water supply and sanitation are the responsibility of local government. In the case of Jakarta, the responsibility falls on the Governor of DKI Jakarta. A local elected assembly approves budgets and any adjustments in tariffs for water supply.

Since 1998, water services in Jakarta have been managed by two private concession companies which serve the western and eastern sides of the city under 25-year public private contracts. These contracts have been renegotiated several times. There are few formal mechanisms for coordination among these actors and the central government's role is limited to largely advisory and financing functions. While this fragmented governance structure may slow the

**Table 2. Selected descriptive statistics.**

| | Mean | Minimum | Maximum |
|---|---|---|---|
| Area (km$^2$) | 2.47 | 0.28 | 12.98 |
| Population | 39,696 | 3038 | 154,003 |
| Piped water access (% population) | 36 | 0 | 100 |
| Septic tank coverage (% population) | 91 | 50 | 100 |
| Toilet access (% population) | 98 | 78 | 100 |
| Diarrhoea (Number of cases/year) | 479 | 0 | 2792 |
| Water security score | 71 | 55 | 80 |

adoption of IUWM policies and regulations at the national level, the lack of coordination may strengthen incentives on the part of the concessionaires to develop IUWM projects using locally available water sources, such as stormwater or greywater, which do not require the cooperation of local governments outside Jakarta and other external parties.

## 4 Results

Table 2 presents selected descriptive statistics for the 260 "kelurahan" or villages of Jakarta.

Figs 1 and 2 show the spatial distribution of access to piped water supply and access to wastewater infrastructure respectively. Fig 3 shows the spatial distribution of composite water security scores. Fig 4 shows a screenshot of the dashboard for one village, Tebet Timur as an example.

For those indicators not available at village level, city-level data is shown in Table 3. Flood infrastructure investment, governance framework and policy framework were discussed qualitatively in Section 3.

Table 4 reports correlation coefficients between variables. These coefficients capture the mutually reinforcing nature of some of the interlinked elements within the water system.

## 5 Discussion

The data suggest that the "kelurahan" or village is a suitable unit size for implementation and monitoring of IUWM interventions in terms of population, with the majority having 10,000–100,000 residents. 15 of the 260 villages have a population below 10,000 and in these cases two or more neighbouring villages with similar characteristics could be clubbed or bundled together.

As expected, coverage of piped water supply is found to vary widely across the city. This is illustrated in Fig 1. It should be noted that the map shows the proportion of population with a connection by area. It therefore captures both the physical extent of the network and the density of connections to the network. A low density of connections may reflect either constraints on the part of the household or utility to secure a connection, or a lack of demand for connections from households in areas where there are alternative water sources.

A second key impact variable, access to toilets, also varies widely across the city. In ten villages, more than 10% of the population have no access to toilets. Nine of these ten villages are located in northern areas of the city. As we would expect, these areas also have below-average septic tank coverage and higher prevalence of diarrhea.

The extremely limited reach of the centralized sewerage network in Jakarta has been noted in previous studies [48], but the data, illustrated in Fig 2, show that household level infrastructure is also limited in some areas, with 35 villages in which more than 20% of the population does not have access to a septic tank. It can be seen that most of the villages with less than 80%

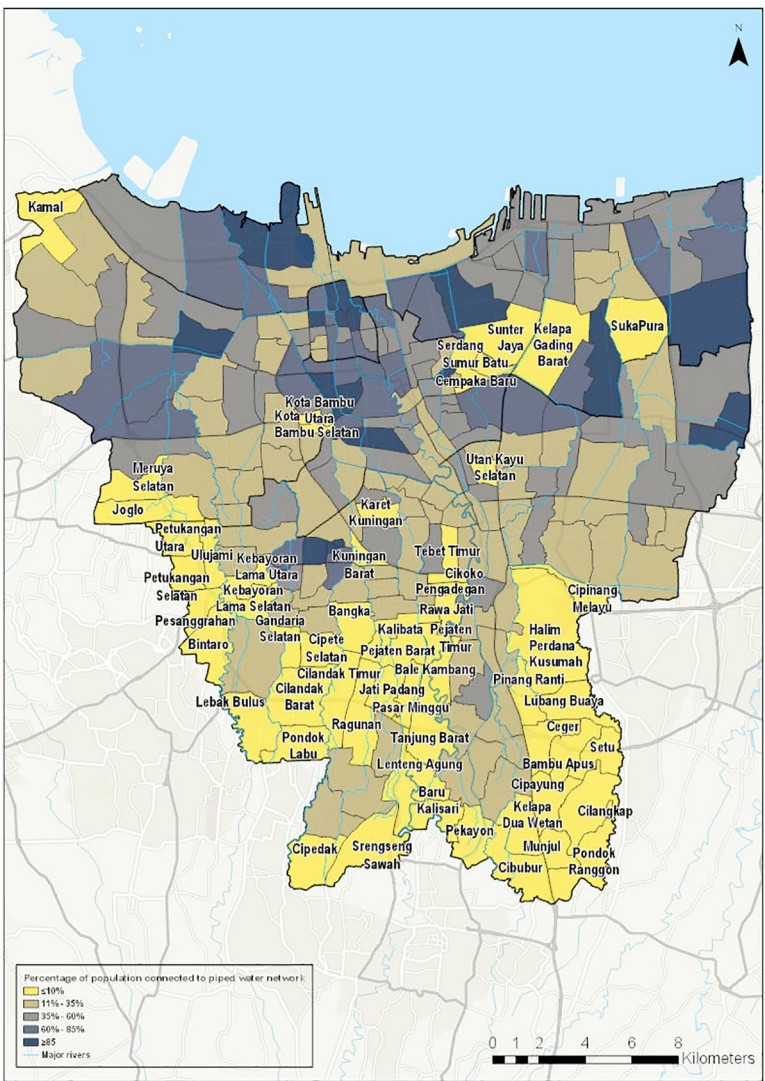

**Fig 1. Access to piped water in Jakarta by kelurahan (village).**

access to a septic tank are located along the lower Ciliwung River, a heavily polluted river [49]. This is consistent with a 2018 study which found that 70–80% of water pollutants in the Ciliwung are from municipal sewage [47], raising risks to health and environmental quality.

These findings may underestimate the health risk posed by inadequate wastewater infrastructure because many septic tanks may be badly installed or poorly maintained and thus are not effective in treating household wastewater. The unsafe disposal of household wastewater is of particular concern in areas of the city where there is a high reliance on groundwater for household use but it is also a concern in areas served by piped supply where low pressure and deteriorated pipe quality may allow infiltration of contaminated groundwater into the tap water distribution network.

Fig 3 shows the spatial distribution of the composite water security score. As such, it brings together the information on hazard, pressure and impact into a single metric which can be used as an initial guide for the prioritization of IUWM projects. There are 36 villages in the lowest score category, corresponding to an overall low level of water security. They are

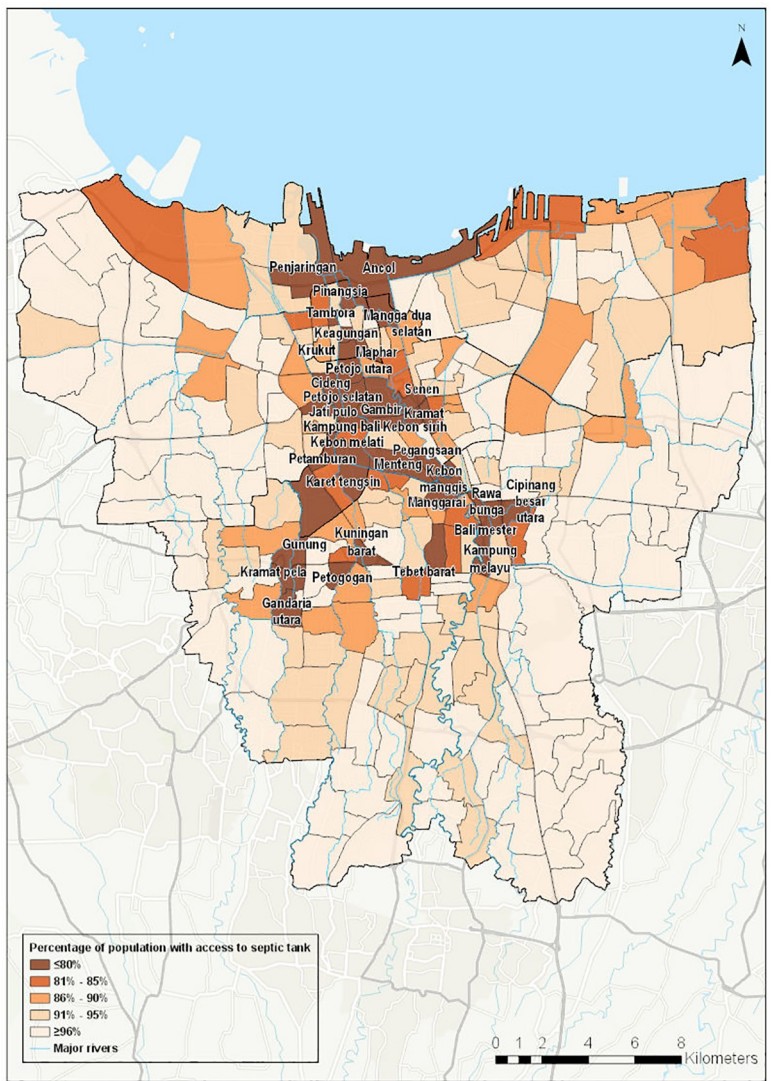

**Fig 2. Access to septic tank in Jakarta kelurahan (village).**

clustered in central Jakarta on the banks of the Ciliwung River, along the northern coast and in the south-west of the city. The central areas are characterised by older, high density housing. Although the piped network extends into these areas, a large proportion of households do not have connections. The coastal villages face underlying pressure from their location at low elevations and exposure to multiple flood types (coastal, riverine and pluvial). Many of these villages also have a higher proportion of slum households. Villages in the south-west of the city are unserved by the piped network, which accounts for the low scores in those areas.

The data allow us to investigate further the relationships between individual indicators to understand the strength of the relationship between components of the urban water system. Fig 5 shows the relationship between piped water coverage and groundwater status. Just over half (53%) of the villages in our low score group are located in areas in which groundwater is classified as damaged or critical. The correlation is moderate (correlation coefficient: -0.45) with lower levels of piped water coverage associated with critical groundwater status (low availability and quality). Thus residents of these areas face a double challenge, as they do not

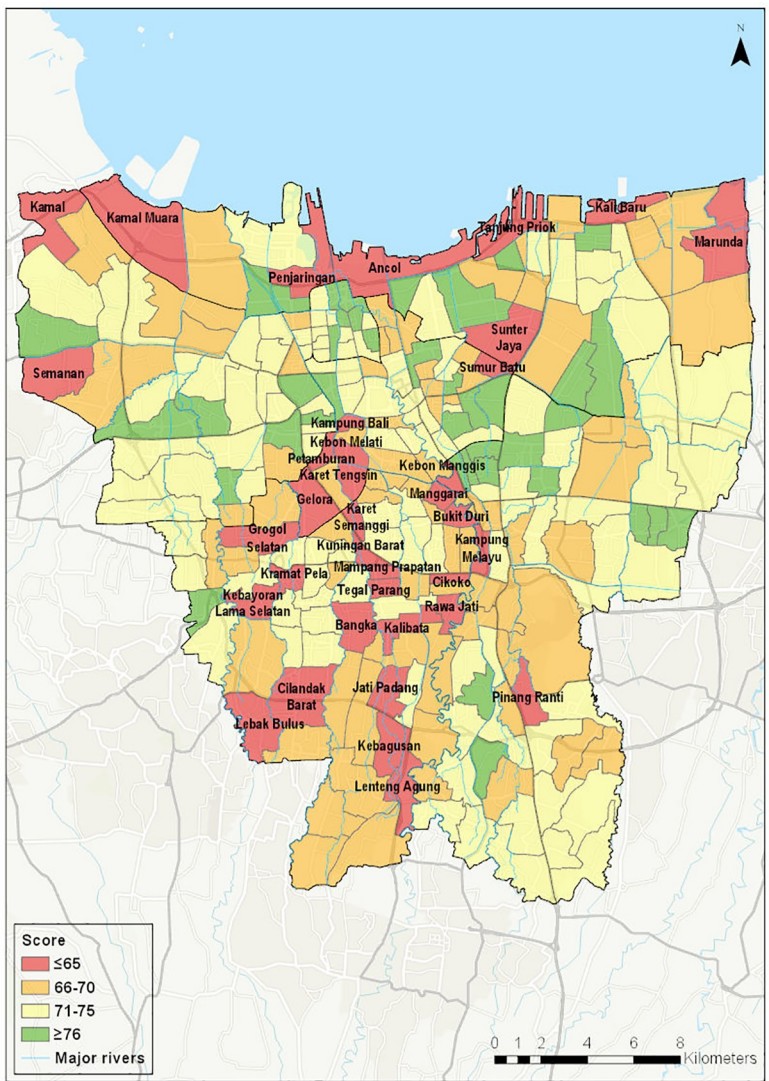

**Fig 3. Aggregated water security score by kelurahan (village).**

have access either to piped water or to safe groundwater, implying an urgent need to develop new sources of safe water supply for residents in these areas. Fig 6 suggests that this situation is likely to deteriorate further in the future as daily groundwater consumption is higher in areas with poor groundwater status (correlation coefficient: 0.33). This may reflect the need for residents to rely on groundwater for water supply even though they may need to sink wells deeper to reach dwindling groundwater reserves.

There is no significant correlation between prevalence of septic tanks and the measure of groundwater status used (Fig 7). As poor groundwater quality is perceived to be a concern in areas with low prevalence of septic tanks and where septic tanks may not be functioning effectively, the absence of correlation may be due to the particular indicator of groundwater status available, which does not include bacteriological contamination.

Figs 8–10 show the relationship between economic activity, poverty and flooding. Flood incidence is a backward-looking measure which captures how many times an area flooded over 2013–2016, the most recent period for which data are available. Fig 8 shows a positive,

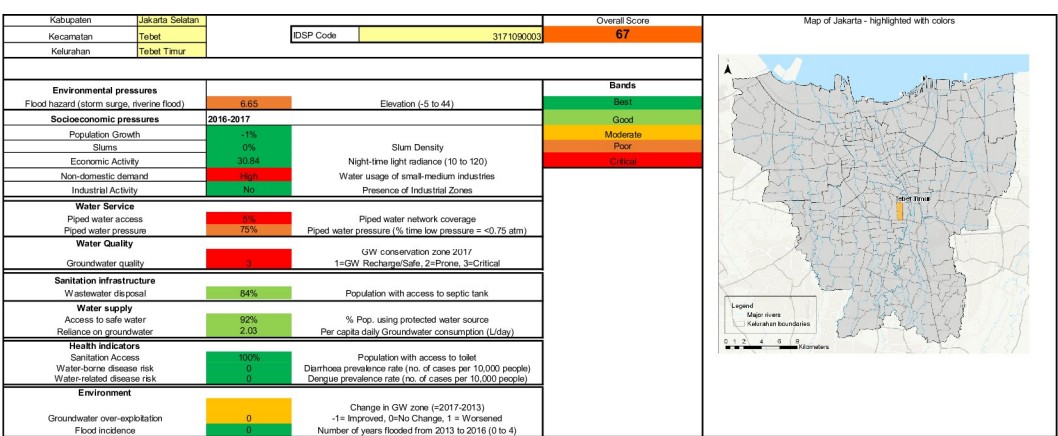

**Fig 4. Water security dashboard presentation example: Kelurahan Tebet Timur.**

**Table 3. City-level indicators.**

| | INDICATOR | UNIT | VALUE |
|---|---|---|---|
| **1101** | Surface water availability (reservoir vol.) | m$^3$ | 234,160[1] |
| **1102** | Precipitation (annual) | mm | 1816 |
| **1103** | Rainfall intensity/variability | mm | ±43–300 |
| **2103** | Affordability | % of average monthly income | 4[1] |
| **2201** | Drinking water quality | % meet standards | 97.5[2] |
| **2400** | Flood protection infrastructure | Qualitative | |
| **4001** | Institutional/governance framework | Qualitative | |
| **4002** | Planning | Qualitative | |

Sources

[1]BPPSPAM (Badan Peningkatan Penyelenggaraan Sistem Penyediaan Air Minum). 2018. Buku Kinerga PDAM 2018: Wilayah II. Jakarta: BPPSPAM. Available at: http://sim.ciptakarya.pu.go.id/bppspam/assets/assets/upload/Wilayah_II_FA.pdf

[2]BRPAMDKI (Badan Regulator Pelayanan Air Minum). 2017. "Kinerja Kuartal I /2017: Tekanan dan Kualitas Air Minum Jakarta". Available at: http://www.brpamdki.org/peformance-2017/detail/190/ Note: Drinking water quality is measured at the outlet of the Water Treatment Plant, not at the tap.

**Table 4. Inter-variable correlation.**

| Indicator 1 | Indicator 2 | Correlation coefficient |
|---|---|---|
| Piped water coverage | Groundwater status | -0.45 |
| Groundwater consumption | Groundwater status | 0.33 |
| Septic tank access | Groundwater status | 0.04 |
| Slum density | Flood incidence | 0.18 |
| Economic activity (radiance) | Flood incidence | -0.31 |
| Economic activity (radiance) | Elevation | -0.53 |

Groundwater status is measured as a 3-way classification: 1 = safe/recharge zone; 2 = prone, 3 = critical/damaged

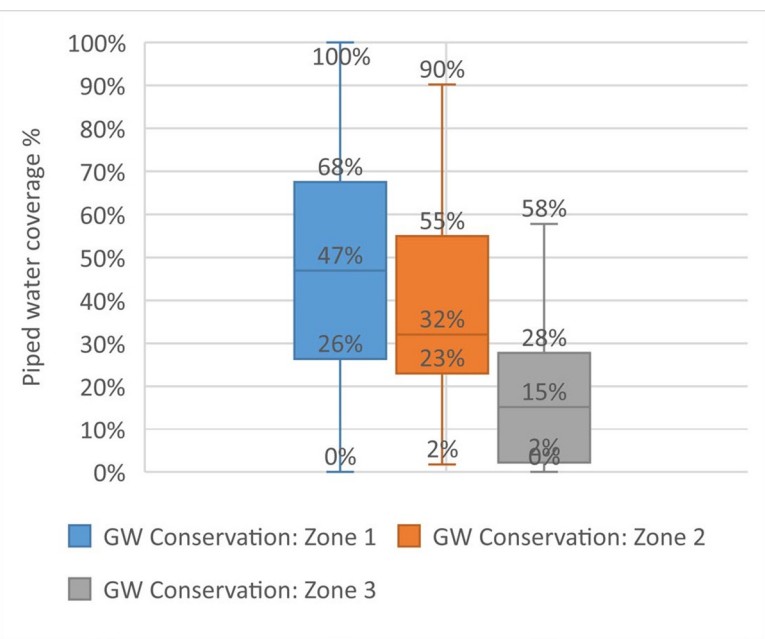

**Fig 5. Piped water coverage and groundwater status.**

moderate-weak correlation (0.18) between the proportion of slum households in the area and flood incidence, reflecting the concentration of low-income households in areas with higher flood risks. This underscores the need to take socioeconomic dimensions of vulnerability into account in flood risk management interventions.

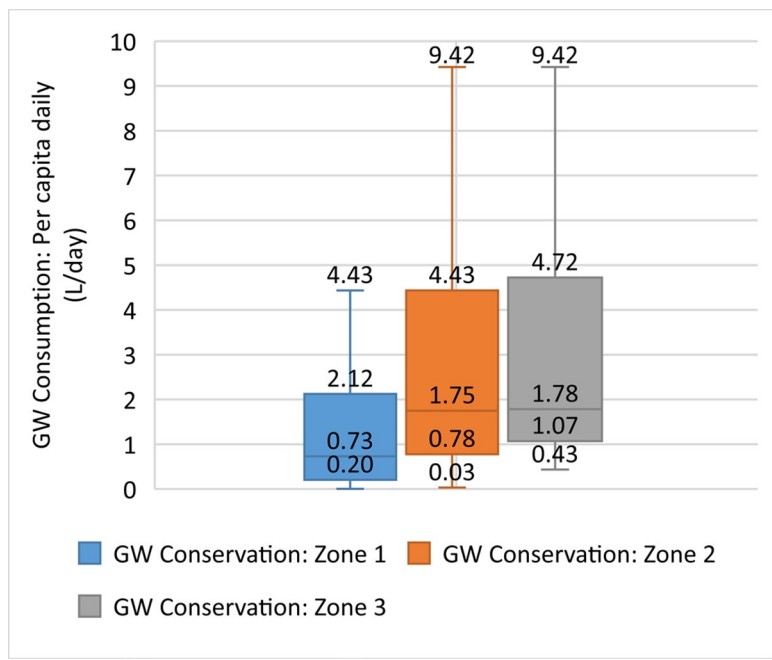

**Fig 6. Groundwater consumption and groundwater status.**

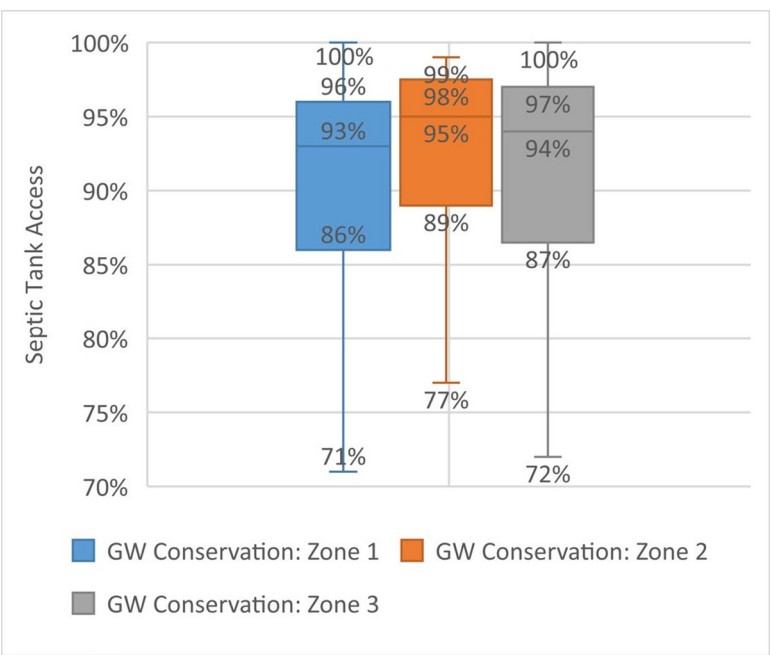

**Fig 7. Septic tank access and groundwater status.**

Fig 9 shows the relationship between economic activity, proxied by radiance, and flood incidence. It shows a moderate negative relationship between the two variables, which may reflect the fact that economic activity has been re-located outside the most flood prone areas or higher levels of flood protection infrastructure have been built in these areas. This may be considered an encouraging finding in terms of the property value at risk from flooding but it may also raise equity concerns if flood defence investment is concentrated in these areas at the

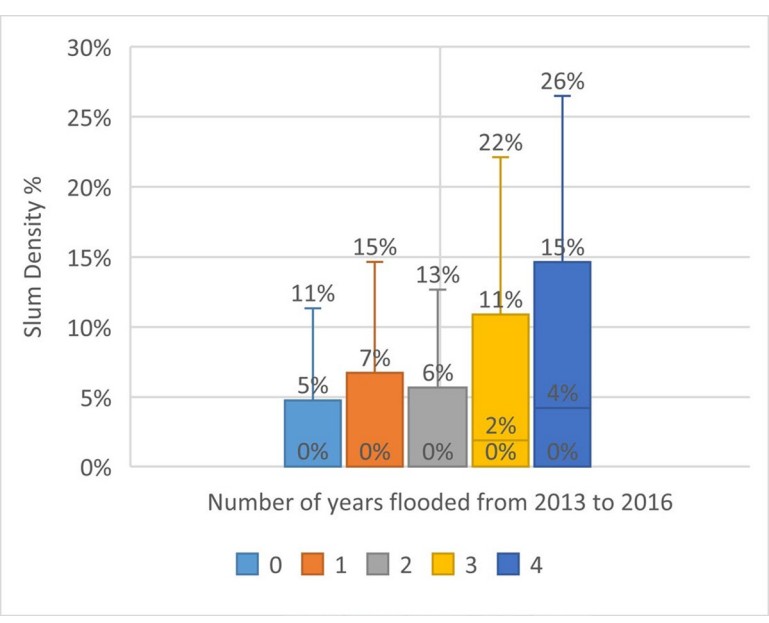

**Fig 8. Slum density and flood incidence.**

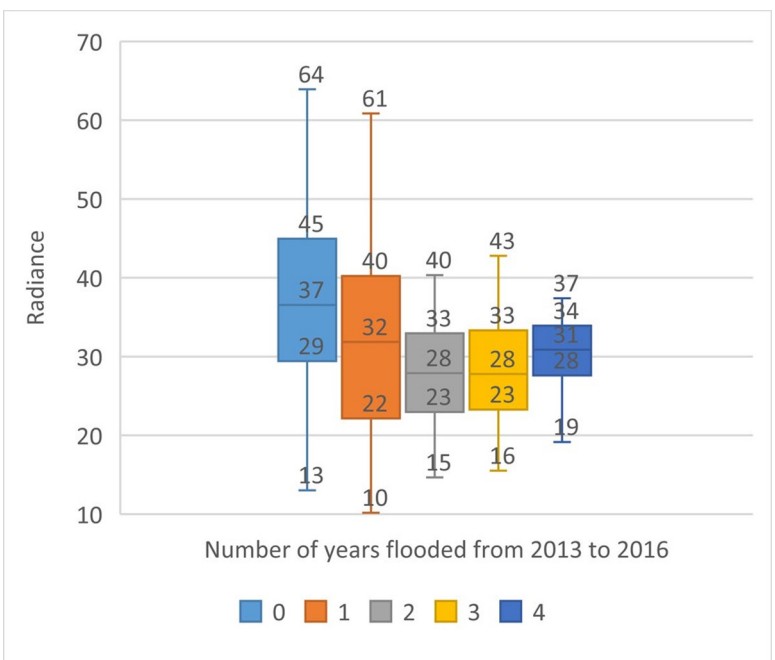

**Fig 9. Radiance and flood incidence.**

expense of flood-prone residential districts. Moreover, Fig 10 shows potential high future economic exposure to flood damage. Using elevation as an indirect proxy of future flood exposure (in the absence of adequate flood protection infrastructure), the figure shows clustering of areas of high economic activity at elevations below sea-level and exposure is likely to increase in the future as a result of continuing land subsidence in the city.

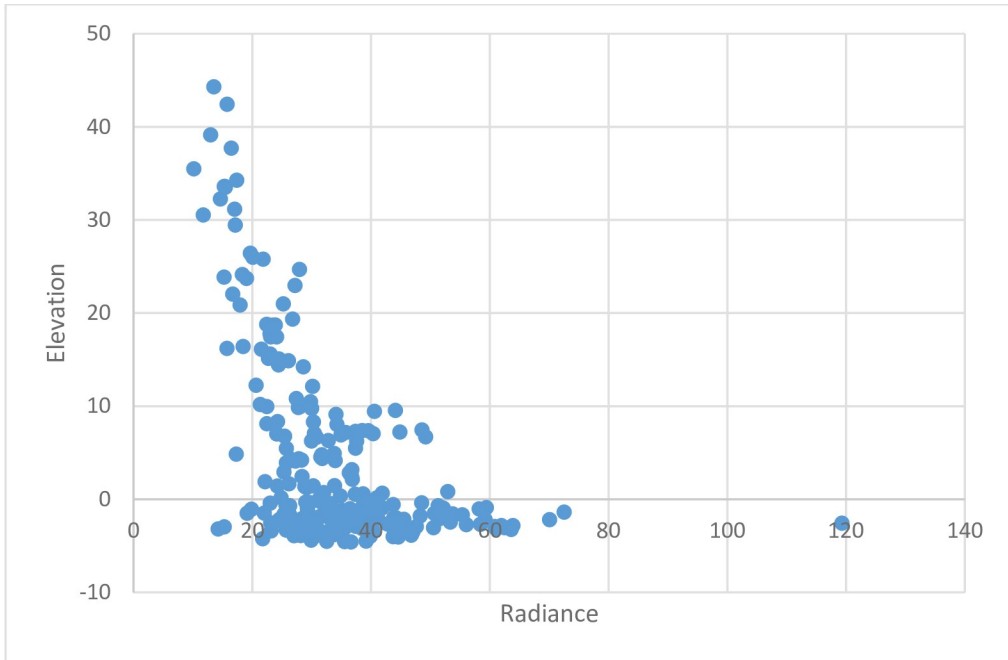

**Fig 10. Radiance and elevation.**

In summary, the data allow us to identify areas with critical groundwater status, high groundwater consumption, low piped supply coverage and low septic tank use which would be suitable priority areas for IUWM interventions. The scores also illustrate the value of micro-level water security analysis as there is variation in each indicator score among villages, such that each village or a cluster of villages may warrant different IUWM interventions.

## 6 Conclusions

The analysis points to the urgent need to develop new sources of water to increase household access to safe water supplies, reduce dependence on low-quality groundwater and control the over-abstraction of groundwater in certain parts of the city. The potential for IUWM interventions to address these challenges was considered by stakeholders at focus group discussions.

Three types of IUWM projects were identified for their applicability in Jakarta: rooftop rainwater harvesting on large buildings; on-site greywater recycling in commercial and industrial facilities; and decentralized small-scale wastewater treatment systems. Rainwater harvesting and on-site recycling were considered to be feasible and beneficial given the existing policy regime, institutional framework and availability of resources, but stakeholders identified a range of regulatory, financial and organizational constraints to the development of these projects. Minimally, regulations should allow for connections to micro-networks to count towards the concessionaires' targets for increasing connections, as long as the quality of the water provided meets drinking water standards.

As the concession contracts come to an end in 2023, there is an opportunity to shape the future governance framework to one which would actively support the adoption of IUWM through targets, incentive schemes, contracts and coordination mechanisms at the municipal level, at higher tiers of government and with financing institutions. Furthermore, Jakarta's water system is heavily influenced by conditions and actions taken upstream and in neighbouring jurisdictions. Mechanisms of oversight and coordination between upstream and downstream local governments are currently weak and will need to be strengthened in order to achieve policy goals efficiently and effectively.

Drawing on and analyzing data from multiple sources, primarily public, suggests that there is value in inter-ministerial or inter-agency collaboration in data-sharing and policy intervention in multi-faceted issue areas like urban water systems. From a micro-level to a transboundary scale, the interaction between ecological, social and economic variables is important in identifying effective IUWM efforts.

In order to refine the analysis further, more precise data on ground and surface water quantity and quality, piped water service quality, subsidence and flood risk would be required. Some of this data has been collected but is held by government agencies, concessionaires and researchers and is not made public. Bringing this data into the public domain could improve policy design and implementation and should be supported by government.

In this paper, IUWM has been understood as an approach and the project types that we have highlighted and explored in initial engagements with stakeholders are suggestive rather than exclusive. The major tasks of setting the scope, technology and arrangements for design and delivery of individual IUWM projects remain to be undertaken. Ideally, the use of IUWM to meet policy objectives in Jakarta can be made more effective through a strong evidence base, while allowing scope for innovation and refinement to meet local needs.

## Supporting information

**S1 File. DKI Jakarta urban water security indicators data details and sources.**
(DOCX)

**S2 File.**
(DOCX)

# Acknowledgments

The authors are grateful for the excellent research assistance provided by Chitranjali Tiwari.

# Author Contributions

**Conceptualization:** Olivia Jensen.

**Data curation:** Olivia Jensen, Adilah Khalis.

**Formal analysis:** Olivia Jensen, Adilah Khalis.

**Funding acquisition:** Olivia Jensen.

**Investigation:** Olivia Jensen, Adilah Khalis.

**Methodology:** Olivia Jensen, Adilah Khalis.

**Project administration:** Olivia Jensen.

**Resources:** Olivia Jensen.

**Software:** Olivia Jensen, Adilah Khalis.

**Supervision:** Olivia Jensen.

**Validation:** Olivia Jensen, Adilah Khalis.

**Visualization:** Olivia Jensen, Adilah Khalis.

**Writing – original draft:** Olivia Jensen, Adilah Khalis.

**Writing – review & editing:** Olivia Jensen, Adilah Khalis.

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
