## [Decision Letter · Decision Letter 0]

5 Nov 2019

PONE-D-19-19667

Urban water systems: development of micro-level indicators to support integrated policy

PLOS ONE

Dear Dr Jensen,

Thank you for submitting your manuscript to PLOS ONE. After careful consideration, we feel that it has merit but does not fully meet PLOS ONE’s publication criteria as it currently stands. Therefore, we invite you to submit a revised version of the manuscript that addresses the points raised during the review process.

We would appreciate receiving your revised manuscript by Dec 20 2019 11:59PM. To enhance the reproducibility of your results, we recommend that if applicable you deposit your laboratory protocols in protocols.io, where a protocol can be assigned its own identifier (DOI) such that it can be cited independently in the future. For instructions see: http://journals.plos.org/plosone/s/submission-guidelines#loc-laboratory-protocols

We look forward to receiving your revised manuscript.

Kind regards,

Monjur Mourshed, Ph.D., B.Arch.

Academic Editor

PLOS ONE

Journal Requirements:

Reviewers' comments:

Reviewer's Responses to Questions

**Comments to the Author**

1. Is the manuscript technically sound, and do the data support the conclusions?

Reviewer #1: Partly

Reviewer #2: Yes

2. Has the statistical analysis been performed appropriately and rigorously? 

Reviewer #1: Yes

Reviewer #2: Yes

3. Have the authors made all data underlying the findings in their manuscript fully available?

Reviewer #1: Yes

Reviewer #2: Yes

4. Is the manuscript presented in an intelligible fashion and written in standard English?

Reviewer #1: Yes

Reviewer #2: Yes

5. Review Comments to the Author

Reviewer #1: The manuscript aspires to provide a set of criteria for prioritizing adoption of integrated urban water management practices in Jakarta, Indonesia. The particular contribution the authors seek to make is two-fold: to first provide a set of indicators of water insecurity/security; and second, to provide hard data, based on locally-available sources of information, to help calibrate the magnitude of these local water security indicators. The effort is worthy of being pursued and refined - and the results could be publishable with additional thought and more rigorous conceptualization of certain challenges - as noted below. However, at this stage the work insufficiently considers important foundational principles needed to generate outcomes useful to decision-makers - and to advancing scholarship on IUWM. There are four major problems with the argument.

First, the authors fail to demonstrate why the mere provision of good data and innformation on water security alone should induce institutional change or adoption of these indicators as reasons for implementing IUWM (e.g., lines 112-115). Given the risk aversion of many water agencies toward innovative measures generally, it may be the case that action to implement IUWM would not be induced through identifying the most urgent areas in a city with major water security problems, but instead, in identifying those areas where programmatic change might be easiest to bring about because change would be less prone to political or public resistance. Similarly, the authors have not effectively made the case that where changes at the city level are difficult to implement, that locality or district implementation of IUWM could be easier (lines 85-87). Why should this be so? What incentives or motivations facilitate confidence that these measures are easier to implement in smaller areas?

Second, given that water supply within Jakarta's otherwise fragmented mangement and governance system for water provision is currently privatized and controlled by two enterprises (lines 280-283), would it not be reasonable to expect some possible resistence to implementation of IUWM measures, particularly if their adoption might affect profit margins and/or managerial control of water sources? A number of studies of this challenge, including at least one on Indonesia, are worth incorporating in this context - see, for example, the following:

1. Birdsall, N. and Nellis, J. (2003) ‘Winners and Losers: Assessing the Distributional Impact of Privatization’, World Development 31 (10): 1617-33.

2. Davis, J. (2004) ‘Corruption in Public Service Delivery: Experience from South Asia’s Water and Sanitation Sector’, World Development 32 (1): 53-71.

3. Al 'Afghani, M. M. (2012), Anti‐Privatisation Debates, Opaque Rules and ‘Privatised’ Water Services Provision: Some Lessons from Indonesia. IDS Bulletin, 43: 21-26. doi:10.1111/j.1759-5436.2012.00303.x

Third, the authors do not enumerate precisely what they mean by IUWM measures and what specific suites of such measures would be appropriate for the tackling of Jakarta's water security challenges (lines 402-406, especially). Would these be measures to better conserve and/or increase the end-use efficiency of potable water use; reuse of wastewater/harvesting stormwater for public use? In the Jakarta context, what might be an IUWM practice, or suite of practices, that have been actively discussed by decision-makers (lines 448-450, for instance)?

Finally, the authors fail to account for recent research which tries to link indicators of water security - in the sense of urgent water problems whose solution is not tractable under current urban water management schemes - with incentives for institutional changes (this could expland tjheir discussion on lines 430- 431, for example). this is something that Peter Gleick, for example, refers to as "predictors of urban water transitions" (Peter H. Gleick, Transitions to freshwater sustainability, PNAS September 4, 2018 115 (36) 8863-8871; https://doi.org/10.1073/pnas.1808893115). Gleick's work could be useful in helping characterize the types of indicator data that would be most useful for inducing such transitions to be pursued by decision-makers.

Finally, there are a number of more minor, but important issues, omitted or skirted in the manuscript that need to be addressed. For example, line 32 should state "these systems" (plural); line 48 should re-state as "centrally-operated distribution and treatment systems." Finally, in the discussion of flood risk as a water security issue (lines 252 and 253) the authors fail to elaborate on why structural measures taken have been insufficient in reducing risk. Is it due to an increase in impervious surface; more land being prone to flooding due to land settling; a greater risk to populations who have chosen to reside in low-lying areas over time; or some combination of these factors? Similarly, on lines 396-7, the econimic exposure to flood damage in some districts may reflect a different phenomenon - that the poorest, most vulnerable, and least economically productive population lives in flood prone districts. This possibility should at least be explored.

Reviewer #2: This paper proposes a set of performance indicators to priorize integrated water projects. A system approached is used for this purpose. The city of Jakarta is used as a case study. Although the topic of the paper is valid to be investigated, the paper suffers from several issues that need improvements. Some of them are the following:

a) The paper needs of a strong revision, including the English language and the elimination of several typos;

b) The indicators proposed, a better justification and a more holistic approach should be adopted. For example, see the paper of Marques et al. (2015) in Environmental Science & Policy. Vol. 54, pp. 142-151.

c) The systems approach adopted needs to be justified. Why not MCDA?

d) The introduction should be improved including a clear description of the objectives, methodology and the contributions for the literature;

e) The conclusions should provide policy implications of the research carried out.

6. PLOS authors have the option to publish the peer review history of their article (what does this mean?). If published, this will include your full peer review and any attached files.

Reviewer #1: Yes: David Lewis Feldman

Reviewer #2: No

---

## [Author Response · Author response to Decision Letter 0]

20 Dec 2019

Responses to Reviewers

PONE-D-19-19667 Urban water systems: development of micro-level indicators to support integrated policy

We are very grateful to the reviewers for their comments and have sought to revise and improve the paper accordingly. Changes are indicated in red in the revised manuscript.

Reviewer #1: The manuscript aspires to provide a set of criteria for prioritizing adoption of integrated urban water management practices in Jakarta, Indonesia. The particular contribution the authors seek to make is two-fold: to first provide a set of indicators of water insecurity/security; and second, to provide hard data, based on locally-available sources of information, to help calibrate the magnitude of these local water security indicators. The effort is worthy of being pursued and refined - and the results could be publishable with additional thought and more rigorous conceptualization of certain challenges - as noted below. However, at this stage the work insufficiently considers important foundational principles needed to generate outcomes useful to decision-makers - and to advancing scholarship on IUWM. There are four major problems with the argument.

First, the authors fail to demonstrate why the mere provision of good data and information on water security alone should induce institutional change or adoption of these indicators as reasons for implementing IUWM (e.g., lines 112-115). Given the risk aversion of many water agencies toward innovative measures generally, it may be the case that action to implement IUWM would not be induced through identifying the most urgent areas in a city with major water security problems, but instead, in identifying those areas where programmatic change might be easiest to bring about because change would be less prone to political or public resistance. 

We agree absolutely that the provision of information by itself does not induce institutional change. Our motivation for focusing on the case of Jakarta for this study is that a confluence of factors has opened a policy space for IUWM there at the present time. 

In particular, the central government has set ambitious targets to expand access to safe water supply and sanitation and has expressed its support for using IUWM approaches to reach the targets. Local governments are responsible for meeting the water supply targets but face severe raw water resource constraints as groundwater abstraction is increasingly strictly regulated and surface waters are declining in quality and fully allocated in the Jakarta metro area. Thus, in Jakarta, the local government, water utility and concessionaires are seeking to identify additional sources of water from within the jurisdiction. 

The shift towards IUWM is backed by international financial institutions, notably the World Bank, which has launched an IUWM initiative, with a view to establishing a financing facility, and by non-governmental organisations such as APEKSI, the organisation of municipal governments. 

We therefore see these indicators as playing a double role, first in helping policy-makers, planners and managers in Jakarta to direct investment to those areas with the greatest need in order to meet government targets effectively and efficiently and secondly to help central government agencies and lenders to monitor progress towards water and sanitation targets. While it is often the case that sustainability indicators are not used by decision-makers (Lehtonen, M. 2013 The non-use and influence of UK energy sector indicators. Ecol. Indic. 35, 24–34.), we hope to increase the likelihood that these indicators will be used by consulting with stakeholders during the development of the indicators and linking them to their policy objectives. 

We have included some additional information on the interactions with stakeholders in the revised version of the paper (Sections 2 & 6).

Similarly, the authors have not effectively made the case that where changes at the city level are difficult to implement, that locality or district implementation of IUWM could be easier (lines 85-87). Why should this be so? What incentives or motivations facilitate confidence that these measures are easier to implement in smaller areas?

The paper did not distinguish clearly between the level at which design and planning of IUWM takes place and the scale of the interventions. In the case of Jakarta, the actors leading on selecting, designing and implementing IUWM projects would be Pam Jaya, the public water utility, and the two private concessionaires, reflecting the allocation of authority under the current governance structure. On the other hand, the IUWM interventions which are under consideration are small-scale interventions like rainwater harvesting, onsite water recycling and distributed micro wastewater collection and treatment systems. These small-scale projects are considered to be more promising for a number of reasons. Firstly, conditions vary widely across Jakarta in terms of the coverage of the water supply network, quality of service (pressure, continuity), groundwater level and quality, flood risk etc. so appropriate project types are likely to vary across localities. Secondly, the existing sewage and drainage systems are very limited, necessitating very high upfront investment to develop centralised reuse using wastewater or stormwater (as applied in Singapore, for example). Thirdly, as IUWM is not yet well established in Indonesia, pilots are needed to demonstrate benefits and costs in the context of a tropical mega-city. We have revised the paper to clarify the role of Pam Jaya and the concessionaires. 

Second, given that water supply within Jakarta's otherwise fragmented management and governance system for water provision is currently privatized and controlled by two enterprises (lines 280-283), would it not be reasonable to expect some possible resistance to implementation of IUWM measures, particularly if their adoption might affect profit margins and/or managerial control of water sources? 

It is correct that the concessionaires might resist IUWM if it led commercial and industrial customers to switch from piped water to onsite harvested or recycled water. However, the private companies have expressed their support for IUWM in principle. This is in part explained by the fact that many of industrial and commercial customers currently rely on groundwater rather than piped water so switching would reduce the pressure on groundwater resources rather than affecting piped water demand. Furthermore, under the contracts the concessionaires receive a fee for each cubic metre of water billed (regardless of the tariff paid by the customer under the tiered tariff structure) and are thus incentivised to increase coverage and to increase continuity of supply and pressure for existing customers. As they have not been able to negotiate an increased allocation from the city’s main external raw water source, expanding supply will only be possible if they can develop new water sources within their jurisdiction. 

It is also relevant to note that the concession contracts will reach the end of their term in 2023 and government parties are currently considering options for the structure of service provision. The future operator is likely to be a public company and will face similar challenges to the concessionaires in expanding supply. IUWM initiatives taken now will allow the government to assess their suitability under new governance arrangements after 2023.

 Third, the authors do not enumerate precisely what they mean by IUWM measures and what specific suites of such measures would be appropriate for the tackling of Jakarta's water security challenges (lines 402-406, especially). Would these be measures to better conserve and/or increase the end-use efficiency of potable water use; reuse of wastewater/harvesting stormwater for public use? In the Jakarta context, what might be an IUWM practice, or suite of practices, that have been actively discussed by decision-makers (lines 448-450, for instance)?

This was the subject of focus group discussions held with stakeholders in Jakarta in October 2019. The project types identified are summarised briefly in Section 6. Our expectation is that the suite of project types considered by stakeholders will evolve and expand over time. 

Finally, the authors fail to account for recent research which tries to link indicators of water security - in the sense of urgent water problems whose solution is not tractable under current urban water management schemes - with incentives for institutional changes (this could expand their discussion on lines 430- 431, for example). this is something that Peter Gleick, for example, refers to as "predictors of urban water transitions" (Peter H. Gleick, Transitions to freshwater sustainability, PNAS September 4, 2018 115 (36) 8863-8871; https://doi.org/10.1073/pnas.1808893115). Gleick's work could be useful in helping characterize the types of indicator data that would be most useful for inducing such transitions to be pursued by decision-makers.

Thank you for pointing us towards Gleick’s work on water transitions which is indeed relevant. The current situation in Jakarta combines several of the conditions identified by Gleick - resource constraints coupled with failures of the existing system leading to over-abstraction of groundwater, subsidence and inadequate provision of safe water supply and sanitation. We have incorporated these points in lines 111-120.  Finally, there are a number of more minor, but important issues, omitted or skirted in the manuscript that need to be addressed. For example, line 32 should state "these systems" (plural); line 48 should re-state as "centrally-operated distribution and treatment systems." 

Corrections made.

Finally, in the discussion of flood risk as a water security issue (lines 252 and 253) the authors fail to elaborate on why structural measures taken have been insufficient in reducing risk. Is it due to an increase in impervious surface; more land being prone to flooding due to land settling; a greater risk to populations who have chosen to reside in low-lying areas over time; or some combination of these factors? 

We have rewritten this section which was misleading. Structural measures – dredging, enlargement of flood water storage capacity – have been effective in reducing flood risk. However, several other factors are increasing flood risk at the same time: changes in upstream land use leading to higher water volumes and velocity, and land subsidence downstream. The measures taken are expected to be inadequate to withstand future pressures. 

Similarly, on lines 396-7, the economic exposure to flood damage in some districts may reflect a different phenomenon - that the poorest, most vulnerable, and least economically productive population lives in flood prone districts. This possibility should at least be explored.

We agree with this point, which is reflected in Figure 8, lines 423-4 and 431-2.

Reviewer #2: This paper proposes a set of performance indicators to prioritize integrated water projects. A system approached is used for this purpose. The city of Jakarta is used as a case study. Although the topic of the paper is valid to be investigated, the paper suffers from several issues that need improvements. Some of them are the following: a) The paper needs of a strong revision, including the English language and the elimination of several typos;

We have reviewed the text thoroughly and have made corrections.

 b) The indicators proposed, a better justification and a more holistic approach should be adopted. For example, see the paper of Marques et al. (2015) in Environmental Science & Policy. Vol. 54, pp. 142-151.

This point is well taken, and we have extended and deepened Section 2 of the paper in order to address this. The initial selection of indicators is drawn from van Ginkel et al 2018. In our view, this set of indicators is holistic as it covers the different facets of water – social, economic and environmental; its different uses; and associated risks. From this larger set, we focus on a subset of indicators which vary at the micro level for the analysis. We are unable to populate all the indicators due to data constraints and use proxies where possible. 

 c) The systems approach adopted needs to be justified. Why not MCDA?

This is an important point and we have sought to express this more clearly in lines 144-150. The systems approach is well suited to capture the interrelated mechanisms of the urban water system and can shed light on relationships of cause and effect between indicators, and thus to inform the selection of interventions. At this stage, we have not sought to prioritize and weight the indicators, focusing instead on the relationships between them. Future work could employ MCDA to address this. 

 d) The introduction should be improved including a clear description of the objectives, methodology and the contributions for the literature;

We have clarified the objectives and contribution in lines 103-109. The selection of the study site is explained in lines 111-120. The approach and method are detailed in Section 2. 

 e) The conclusions should provide policy implications of the research carried out.

We have revised the concluding section (Section 6) to draw out policy implications and directions for further research. Following the initial submission of this paper, a stakeholder workshop was held in Jakarta to discuss suitable IUWM models and to discuss the opportunities and constraints for IUWM and the findings are reported briefly in the revised version. Future work will focus on identifying barriers to the development of these IUWM project types in the priority locations identified.

---

## [Decision Letter · Decision Letter 1]

13 Jan 2020

Urban water systems: development of micro-level indicators to support integrated policy

PONE-D-19-19667R1

Dear Dr. Jensen,

We are pleased to inform you that your manuscript has been judged scientifically suitable for publication and will be formally accepted for publication once it complies with all outstanding technical requirements.

With kind regards,

Monjur Mourshed, Ph.D., B.Arch.

Academic Editor

PLOS ONE

Additional Editor Comments (optional):

Reviewers' comments:

Reviewer's Responses to Questions

**Comments to the Author**

1. If the authors have adequately addressed your comments raised in a previous round of review and you feel that this manuscript is now acceptable for publication, you may indicate that here to bypass the “Comments to the Author” section, enter your conflict of interest statement in the “Confidential to Editor” section, and submit your "Accept" recommendation.

Reviewer #1: All comments have been addressed

Reviewer #2: All comments have been addressed

2. Is the manuscript technically sound, and do the data support the conclusions?

Reviewer #1: Yes

Reviewer #2: Yes

3. Has the statistical analysis been performed appropriately and rigorously? 

Reviewer #1: Yes

Reviewer #2: Yes

4. Have the authors made all data underlying the findings in their manuscript fully available?

Reviewer #1: Yes

Reviewer #2: Yes

5. Is the manuscript presented in an intelligible fashion and written in standard English?

Reviewer #1: Yes

Reviewer #2: Yes

6. Review Comments to the Author

Reviewer #1: The authors have effectively responded to all my comments on the initial draft submission and the manuscript is much improved.

Reviewer #2: The paper is much better now and I recommend its acceptance. Congratulations and good lucky to the authors.

7. PLOS authors have the option to publish the peer review history of their article (what does this mean?). If published, this will include your full peer review and any attached files.

Reviewer #1: Yes: David L Feldman

Reviewer #2: No

---

## [Editor Report · Acceptance letter]

7 Feb 2020

PONE-D-19-19667R1 

Urban water systems: development of micro-level indicators to support integrated policy 

Dear Dr. Jensen:

I am pleased to inform you that your manuscript has been deemed suitable for publication in PLOS ONE. Congratulations! Your manuscript is now with our production department. 

With kind regards,

on behalf of

Prof. Monjur Mourshed 

Academic Editor

PLOS ONE